# Asymmetric Cellulose/Carbon Nanotubes Membrane with Interconnected Pores Fabricated by Droplet Method for Solar-Driven Interfacial Evaporation and Desalination

**DOI:** 10.3390/membranes12040369

**Published:** 2022-03-29

**Authors:** Zhiyu Yang, Linlin Zang, Tianwei Dou, Yajing Xin, Yanhong Zhang, Dongyu Zhao, Liguo Sun

**Affiliations:** 1School of Chemical Engineering and Materials, Heilongjiang University, Harbin 150080, China; yangzy818yzy@163.com (Z.Y.); doutw1992@126.com (T.D.); xinjingya@163.com (Y.X.); sunliguo1975@163.com (L.S.); 2School of Environmental Science and Engineering, Southern University of Science and Technology, Shenzhen 518055, China; zangll423@163.com

**Keywords:** droplet method, carbon nanotubes, porous materials, interfacial evaporation, solar energy materials

## Abstract

Solar-driven interfacial water purification and desalination have attracted much attention in environmentally friendly water treatment field. The structure design of the photothermal materials is still a critical factor to improve the evaporation performance such as evaporation rate and energy conversion efficiency. Herein, an asymmetric cellulose/carbon nanotubes membrane was designed as the photothermal membrane via a modified droplet method. Under 1 sun irradiation, the evaporation rate and energy efficiency of pure water can reach up to 1.6 kg m^−2^ h^−1^ and 89%, respectively. Moreover, stable reusability and desalination performance made the cellulose/carbon nanotubes membrane a promising photothermal membrane which can be used for solar-driven desalination.

## 1. Introduction

In recent years, solar-driven interfacial evaporation has attracted widespread attention [1,2,3,4]. Many studies have shown that this emerging water treatment technology has great application prospects in seawater desalination and water purification [5,6,7,8]. Compared with traditional desalination technologies such as reverse osmosis (RO), multi-stage flash distillation (MSF), multi-effect distillation (MED) and vapor compression distillation (VCD), solar-driven interfacial desalination can maximize water recovery and reduce fossil fuel consumption by inducing various materials with light-to-heat conversion properties [7,9,10,11]. Therefore, it is considered to be one of the most promising technologies for water purification and desalination [12].

To improve the overall evaporation performance, photothermal membranes with vertically oriented, wrinkled or layered structure can be constructed by freeze-drying and hydrothermal methods to enhance the light absorption capacity and accelerate the vapor diffusion [13,14,15,16,17,18,19]. Although these membranes of different structure can achieve great performance of photothermal evaporation, their photothermal-vapor conversion capacities are relatively low, because the efficient photothermal steam generation is also somewhat related to the porous structure. Solar-driven interfacial desalination used porous structure membranes exhibits more excellent performance in terms of optical absorption, photothermal conversion and photothermal interfacial evaporation [20,21]. At present, the reported preparation methods of porous membranes include template method, stretching and electrospinning [22,23,24]. These preparation methods are expected to be great strategies for preparing photothermal membranes with excellent interfacial evaporation performance, but most of them are complicated in operation, which are not conductive to large-scale application [25,26,27,28]. Therefore, a simple method to synthesize photothermal evaporation membranes with abundant pore structures is urgently needed.

It has been reported that utilizing the gas–liquid interface can fabricate the ordered porous membranes. For example, the porous poly (lactic acid) membrane obtained by polymerizing or precipitating at the gas–liquid interface using the template of monolayer colloidal crystal floating on the liquid surface [29]. Adopting the two-phase anisotropy at the gas–liquid interface, asymmetric membranes with porous structures can be obtained [30,31,32,33,34].

Recently, inspired by membrane formation on the gas–liquid interface, droplet methods have been adopted in preparing porous membranes due to its uncomplicated operation. The typical process is summarized as follows: (1) Preparation of polymer solution for droplets formation. The droplets are homogenous solution composed of polymer, solvent and additions which improve the structure and performance of membranes. (2) Preparation of coagulation bath. The coagulation bath is composed of nonsolvent and solvent. Changing the proportion of solvent and nonsolvent could affect the structure of the prepared membranes. (3) Droplets dropping from the fixed nozzle. The droplets drop from nozzle and fall into the coagulation bath at a certain height to form membrane. There are some factors affecting the formation of membranes such as solution concentration, droplet height [35,36,37]. However, the key point is the selection of solvent and nonsolvent. The solvent must completely dissolve the polymer while the nonsolvent cannot dissolve the polymer, and the solvent should be miscible with the nonsolvent. When the droplet of homogenous solution is in contact with coagulation bath, the exchange of solvent and nonsolvent at the gas–liquid interface leads to phase separation, forming a polymer-rich phase (to form dense skin layer) and polymer-poor phase (to form porous structure). Subsequently, phase separation continues until the polymer-rich phase completely solidifies, forming an asymmetric porous membrane [38,39,40,41].

Herein, an asymmetric cellulose/carbon nanotubes membrane with porous structure was prepared by modified droplet, in which a mixed polymer solution of cellulose, polymethacrylic acid (PMAA), ionic liquid and carbon nanotubes (CNTs) was dropped into the coagulation bath. Under the combined effects of gravity, buoyancy and surface tension, droplet of polymer solution dropping into the coagulation bath would form a membrane at the gas–liquid interface of the coagulation bath. PMAA increased hydrophilicity as a polymer electrolyte and ionic liquid was used to dissolve cellulose. CNTs were used as photothermal materials to convert the absorbed light energy [42,43]. At the same time, it also acts as a supporting network together with cellulose to stabilize the porous structure. The three-dimensional (3D) interconnected pores structure of the membrane can effectively improve the evaporation performance by reducing light reflection and accelerating vapor escape.

## 2. Materials and Methods

### 2.1. Materials

Carbon nanotube were multi-walled CNT with a length of 1~2 μm and outer diameter of 20~40 nm obtained from Shenzhen Nanotech Port Co., Ltd. Cellulose was received from Shanghai Aladdin Bio-Chem Technology Co., LTD. (Shangai, China) 1-Butyl-3-methylimidazolium chloride ([Bmim] Cl, >99%) was obtained from Shanghai Yiji Industrial Co., Ltd. (Shangai, China) Methacrylic acid (MMA, AR), dimethylformamide (DMA, AR), sodium chloride (NaCl, AR), methylene blue (biological dye, BS), methyl orange (biological dye, BS) were purchased from Sinopharm Chemical Reagent Co., Ltd. (Shangai, China). Distilled water was received from Harbin Wenjing Distilled Water Factory.

### 2.2. Acidification of Carbon Nanotubes

First, 5 g pristine multi -walled CNTs with lengths of 1~2 μm were added to 500 mL of the mixed strong acid solution with H_2_SO_4_/HNO_3_ which was a volume ratio of 1/3. The mixture was uniformly mixed under magnetic stirring, then heated to 333 K and stirred at reflux for 3 h. After cooling to room temperature, the solution was poured into a beaker and diluted with deionized water, and then filtered under reduced pressure until the sample became neutral. Finally, the acidified carbon nanotubes were prepared by drying at 323 K in a vacuum drying oven. Through the modification of acidification, oxygen-containing functional groups were added to improve the hydrophilicity of materials so that they can be fully dispersed in solvents.

### 2.3. Fabrication of Cellulose/CNTs Membrane

In this experiment, the cellulose/CNTs membrane was prepared by a droplet method. First, 0.15 g cellulose powder and 0.15 g PMAA were added to 3 g 1-Butyl-3-methylimidazolium chloride ([Bmim] Cl) at 80 °C and stirred for 2 h until they were completely dissolved. Then, 5 mL of N, N-Dimethylformamide (DMF) solution containing 20 mg acidified CNTs was poured into the above solution and mixed thoroughly. After that, the obtained solution was transferred to a 10 mL syringe and dropped into a coagulation bath with DMF/water (volume ratio = 1/1) through a micro-syringe pump at a height of 10 cm above the liquid level, thereby forming a composite membrane on the surface of the coagulation bath with DMF/water solution (Figure 1). Subsequently, the composite membrane was repeatedly washed with water to remove [Bmin] Cl and PMAA, and then freeze-dried using liquid nitrogen. The resulting cellulose/CNTs membrane was named as CCM. As a control, cellulose membrane without any CNTs were also prepared according to the above procedure, which was named as CM. Cellulose membrane without porous structure by freeze-drying prepared solution directly for another control experiment was named as CCM-N.

### 2.4. Characterization

The surface morphologies and cross-section morphologies of the CCM were characterized by a scanning electron microscope (Hitachi, Tokyo, Japan, S-4800). Surface chemical composition of CCM was examined by an ATR-FTIR spectra which was measured by using a Spectrum One instrument (Perkin Elmer, Waltham, MA, USA). UV-vis-NIR diffuse reflectance spectra (DRS) was measured with a Perkin-Elmer Lambda 950 UV-vis-NIR spectrophotometer (USA). The concentrations of ions were detected by inductively coupled plasma-optical emission spectrometer (ICP-OES, Optima 8300, Perkin Elmer, USA).

### 2.5. Solar-Driven Interfacial Evaporation Experiments

To investigate the solar-driven interfacial evaporation performance, the cellulose membrane with/without CNTs was employed as the photothermal material and placed on a transporter-assisted evaporation system. A polystyrene foam and a glass fiber filter were used as the heat insulation and water channel, respectively. In this experiment, the area of the membrane used was 4 cm^2^, and the volume of water used was 20 mL. All evaporation experiments were conducted under a solar simulator (Perfect Light PLS-SXE300DUV). The mass of the water loss is measured by an electrical balance. The *surface temperature* of the photothermal membrane was measured by an infrared thermal image (FLIR ONE). The solar-heat energy conversion efficiency (*η*) was calculated using the following formulas:$$\eta =\frac{\dot{\left(m_{light}-m_{dark}\right)}h_{LV}}{I}\times 100\%$$
$$h_{LV}=c(T_{surface}-T_{0})+L$$
$$L=-0.00006T_{surface}^{3}+0.0016T_{surface}^{2}-2.36T_{surface}^{1}+2500.8$$
where $m_{dark}$ and $m_{light}$ are water evaporation rates under *dark* and *light* conditions, respectively. *h_LV_* is the total enthalpy, *c* is the specific heat capacity and *L* is the specific latent heat of phase change. *T_surface_* and *T*_0_ represented the temperature of the evaporation surface and bulk water reservoir, respectively.

The desalination efficiencies of different ions(*η_i_*) were calculated using the following formulas [44,45,46]:$$\eta_{i}=\frac{C_{0}-C_{t}}{C_{0}}\times 100\%$$
where $C_{0}$ is the concentrations of different ions before desalination and $C_{t}$ is the concentrations of different ions after desalination.

## 3. Results and Discussion

When the droplet enters the DMF/water solution of coagulation bath, they will sink to the liquid under gravity. Meanwhile, [Bmim] Cl molecules in the droplet diffuses into the DMF/water solution, resulting in generating large number of tiny pores. As [Bmim] Cl content in the droplet decreases, the 3D spherical droplet becomes a two-dimensional (2D) membrane and then floats on the surface of the coagulation bath. Since [Bmim] Cl molecules diffuse relatively rapidly when the droplet contacts with DMF/water solution, the cellulose/CNTs mixed polymer solution occurred gelation, dense skin layers are formed on the liquid–liquid surfaces of the CCM (Figure 1). The dense skin layer could help resist the pulling effect of surface tension in the coagulation bath. In contrast, the interior of the CCM exhibited an obvious 3D interconnected porous structure, whose reason lied in the sufficient solvent diffusion.

The modified droplet method for preparing the CCM mainly involves phase-inversion process in the coagulation bath. There are many influencing factors of phase-inversion, and the key factors of phase-inversion are solvent type and composition, polymer type and composition, non-solvent type and composition and membrane forming conditions. In this process, cellulose/CNTs in the droplet is selected as polymer, [Bmin]Cl/DMF in the droplet as solvent, and DMF/water as nonsolvent (the coagulation medium) [47]. Therefore, attention also needs to be paid to the concentration of solvent and nonsolvent. When the droplet enters the coagulation bath, a portion of the droplet surface is in contact with the nonsolvent of the coagulation bath. Due to the gradients in density and/or interfacial energy of the polymer-nonsolvent interface, slow convective flow of nonsolvent-solvent occurs in the droplet, resulting in a large number of small pores which is the sponge-like 3D interconnected porous structure [47,48]. The membrane at liquid–liquid interface presents a dense structure due to the direct contact between the coagulation bath and the droplet. The other part of the droplet surface, that is not in direct contact with the nonsolvent of the coagulation bath, is relatively distant from the nonsolvent, so it has difficulty forming convective flow with the nonsolvent. The relatively high concentration of polymer causes it gather together, resulting in the size of pores at the gas–liquid interface are relatively large [48,49]. The modified droplet method is achieved at the interface by gravity, buoyancy and interaction between the surface tension of cellulose/CNTs mixed solution and the surface tension of the coagulating bath. The main purpose of adding DMF is to adjust the surface tension of cellulose/CNTs mixed polymer solution and coagulation bath. The addition of DMF had little effect on gravity and buoyancy, so their impact could be almost ignored in this research. When the appropriate content of DMF is added to cellulose/CNTs mixed solution, the concentration of cellulose/CNTs mixed solution at the liquid–liquid surface is higher than that at other parts of the membrane through phase separation, so that the CCM would form a dense layer at the liquid–liquid surface. The dense layer can better resist the surface tension of the coagulating bath and prevent the film from being damaged by the surface tension of the coagulating bath. Therefore, when the droplet is dropped into the coagulation bath, the solution bead is formed in the coagulation bath and, at the same time, the original stable state should be destroyed to make the cellulose/CNTs solution flow so as to make the droplet form a membrane [50]. So, there are two crucial factors affecting the preparation of CCM: cellulose/CNTs mixed polymer solution concentration and coagulation bath concentration.

As shown in Figure 2, when the DMF/water volume ratio of the coagulated bath remained unchangeable and the DMF content of droplet increased, the concentration of cellulose/CNTs mixed polymer solution decreased and the surface tension of droplet would decrease, which would also affect the state of CCM formation. It was taken as the research object that DMF/water volume ratio of the coagulation bath was 1/3 (surface tension= 0.07100 N m^−1^). As the DMF content in the droplet increased and concentration of cellulose/CNTs mixed polymer solution decreased, the final state of the droplet in the coagulation bath gradually changed from bead-like to membrane-like and then even was split into small pieces. This was because the surface tension of the droplet was unable to resist the surface tension of the coagulating bath, therefore the droplet was split under the action of the surface tension of the coagulating bath. As can be seen from the Figure 2, due to the surface tension of the cellulose/CNTs mixed polymer droplet which added 4 mL DMF (surface tension = 0.07112 N m^−1^) was too large to form a membrane, the droplet fell into the coagulation bath and formed like-bead float on the coagulation bath under combination of gravity, buoyancy and surface tensions. Meanwhile, the surface tension of the droplet was too small, due to the addition of 7 mL DMF (surface tension = 0.06711 N m^−1^). When the droplet dropped into the coagulating bath, the surface tension of the droplet cannot compete with the surface tension of the coagulating bath. So, it was difficult to form a complete circular membrane and will be dispersed into small pieces or the membrane formed was too thin. With the increase of DMF content, the gelation degree of cellulose/CNTs mixed polymer solution also decreased, and it cannot resist the surface tension of coagulation bath effectively. 

In Figure 2, by fixing DMF content of cellulose/CNTs mixed polymer droplet and increasing DMF/water volume ratio of coagulation bath, the surface tension of coagulation bath would decrease, but the decrement was small. If DMF cellulose/CNTs mixed polymer solution added 6 mL DMF was taken as the research object (surface tension= 0.06756 N m^−1^), we can clearly see that different volume ratio of coagulation bath had different effect on the formation of membrane. With the decrease of the volume ratio, the formed membrane gradually changed from a thin membrane which was split easily to a circular membrane with uniform and appropriate thickness, finally a complete film cannot be formed. Considering the two variables, the membrane which was prepared from cellulose/CNTs mixed polymer solution added 6 mL DMF and water (surface tension = 0.07222 N m^−1^) was similar to the membrane which was prepared from cellulose/CNTs mixed polymer solution added 7 mL DMF (surface tension = 0.06711 N m^−1^) and coagulation bath with DMF/water (volume ratio = 1/3). The appearance further illustrated that CCM obtained from the interaction between the surface tension of coagulation bath and the surface tension of polymer solution. From the above analysis, it was clear that when the DMF content added to cellulose/CNTs polymer solution and DMF/water volume ratio of coagulation bath were appropriate, a circular membrane with uniform and appropriate thickness would be formed. In a word, the coagulation bath with DMF/water (volume ratio= 1/1) and the cellulose/CNTs mixed polymer solution added 6 mL DMF were the most suitable conditions to prepare membranes by the droplet method.

In the coagulation bath with DMF/water (volume ratio = 2/1), less water can be gelatinized for cellulose/CNTs mixed polymer solution, the degree of gelation was lower, while more polymer solution appeared as liquid state compared to others at the same time. When the membrane area reached its maximum, non-gelatinization polymer solution returned to the droplet shape by its own surface tension, resulting in the contraction of membrane as shown in Figure 2.

In Figure 3, the digital photographs taken by the high-speed camera showed that the entire process from droplet to the CCM took 4 ms. In this process, the cellulose/CNTs mixed polymer solution was diffused and formed a complete circular membrane under the interaction of droplet surface tension and coagulation bath surface tension. At 20 ms, a cavity could be seen forming after the droplets had been dropped into the coagulation bath. Then a circular membrane was formed under the interaction of the surface tensions. In less than 0.04 s, CCM could be formed. When a complete circular film was formed, the membrane floated on the coagulation bath under the action of surface tensions. When a complete circular membrane was formed, the membrane floated on the coagulation bath by the action of surface tensions.

A mathematical model was established for analyzing membrane formation through modified droplet method. The A is the spreading coefficient of polymer solution, which is the ratio of spreading ability (*δ*_spread_) to anti-spreading ability (*δ*_anti-spread_). The spreading ability of the polymer solution is derived from the surface tension of the coagulation bath. The spreading resistance (*B*_gel_) is the tensile resistance of the instantaneous gelation product of polymer solution, which is related to the concentration of polymer solution. Meanwhile, gelation is accomplished by non-solvent in the coagulation bath. The ratio of water in the coagulation bath affects the thickness and strength of the gel layer, so k water is induced. The value of A can be used to determine whether the gel can be formed and the state of the membrane.
$$A=\frac{\delta_{\mathrm{spread}}}{\delta_{\mathrm{anti-spread}}}=\frac{\sigma_{\mathrm{cb}}}{k\times B_{\mathrm{gel}}}$$

By the measurement of the gel membrane size, the formula is obtained by fitting. Different requirements of the gel membrane can be prepared by changing the parameters.
$$r_{\mathrm{membrane}}=A\times \frac{a}{\sigma_{\mathrm{polymer}\;\mathrm{solution}}}+b$$
$$h_{\mathrm{membrane}}=\frac{V}{A}=\frac{V}{\pi r_{\mathrm{membrane}}^{2}}$$

We can see from the above research, the best CCM was prepared from cellulose/CNTs mixed polymer solution added 6 mL DMF and coagulation bath with DMF/water (volume ratio = 1/1). Therefore, we chose this membrane for further tests.

SEM images in Figure 4 showed the surfaces and internal morphology of the CCM with a thickness of 400 ± 25 μm. The characteristic of asymmetric membrane can be clearly seen by Figure 4. The dense skin layer which located at liquid–liquid surface was 1~3 μm (Figure 4c). At the gas–liquid interface, pore diameters could up to 22 µm (Figure 4b). After the sufficient solvent diffusion, the interior of the CCM exhibited an obvious 3D interconnected pores structure (Figure 4c,d).

The FTIR spectra shown in Figure 5a demonstrated that the CCM exhibited characteristic peaks at 1708 cm^−1^ and 1053 cm^−1^, which were attributed to the vibration of C=O and C-O-C pyranoid ring of PMAA and cellulose, respectively. Comparing with the cellulose and PMAA samples, a new characteristic peak appeared at 1562 cm^−1^ corresponding to the stretching vibration of COO^−^. Besides, the carboxyl group in PMAA could generate electrostatic attraction with the hydroxyl group in cellulose to form a cross-linked network, which destroyed the intermolecular hydrogen bond of cellulose, with the characteristic peak at 3600–3000cm ^−1^ strength decreased. Comparing with the CM, the CCM exhibited small optical reflectance (≈4–6%) in the 250–2500 nm wavelength range, indicating the large optical absorption of the CCM (Figure 5b).

It was shown in Figure 6 that the transporter-assisted interfacial evaporation system, where the CCM was used as the photothermal membrane to absorb light heat water and generate vapor. As a result, compared with the bulk water and cellulose membrane, the surface temperature of the CCM can quickly rise to 44.4 °C within 600 s under 1 sun irradiation (Figure 7a). Additionally, excellent light absorption and heat localization of the CCM enabled the evaporation system to present higher water evaporation rates under dark and light conditions. As shown in Figure 7b, the membrane with CNTs exhibited an excellent evaporation rate of roughly 1.6 kg m^−2^ h^−1^, and the corresponding light-to-heat energy conversion efficiency of the transporter-assisted evaporation system was calculated to be 89%, which was higher than that of reported photothermal membranes. In contrast, the evaporation rate of the evaporation system using the CM was less than 0.9 kg m^−2^ h^−1^ due to the absence of light absorber and lower surface temperature, and the evaporation rate of the evaporation system using the CMM-N was less than 1.3 kg m^−2^ h^−1^ due to the absence of 3D interconnected pore structure. The results indicated that the addition of CNTs and optimal porous structure can significantly enhance the solar-driven interfacial evaporation performance and maximize the energy conversion efficiency.

We also evaluated the desalination performance of the transporter-assisted evaporation system. Figure 8a showed that evaporation rate of seawater was slightly lower than that of pure water, which was caused by more complex components such as salt ions, natural organic matters and bacteria [51]. For NaCl solutions with different salinities, the evaporation rate of 20 wt% NaCl solution can still reach about 1.4 kg m^−2^ h^−1^ after one-hour evaporation test, although the lower partial vapor pressure resulted in a slower evaporation rate as the concentration increased from 1.4 wt% to 20 wt% (Figure 8a,b). In the 10-cycle test, the evaporator can maintain a stable evaporation rate of about 1.55 kg m^−2^ h^−1^ (Figure 8c). More importantly, during the evaporation of seawater, the desalination efficiencies of Na^+^, Mg^2+^, K^+^ and Ca^2+^ were 99.96%, 99.97%, 99.30% and 99.86%. Meanwhile, the concentrations of Na^+^, Mg^2+^, K^+^ and Ca^2+^ in the condensed water were greatly reduced, which were lower than the salinity levels defined by World Health Organization (WHO) (Figure 8d).

## 4. Conclusions

The asymmetric CCM was fabricated by droplet method with 3D interconnected pores structure and good light absorption ability, which resulted in high evaporation rate and energy conversion efficiency when processing pure water, salt water and fresh seawater under one sun irradiation. The stable reusability and high-quality condensed water made the transporter-assisted evaporation system a potential candidate for solar-driven water purification and desalination.

## Figures and Tables

**Figure 1 membranes-12-00369-f001:**
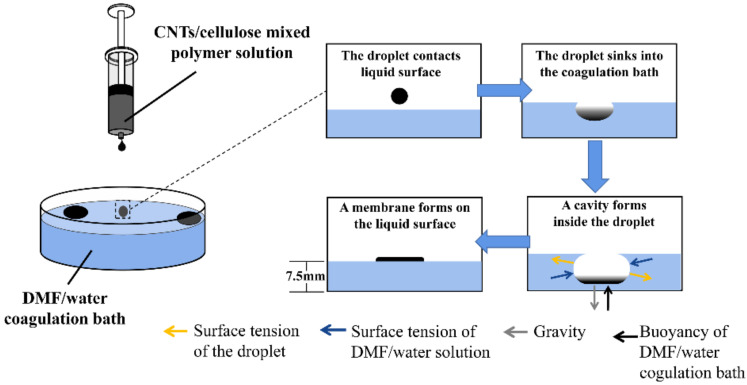
Schematic diagram of the preparation of the cellulose/CNTs membrane.

**Figure 2 membranes-12-00369-f002:**
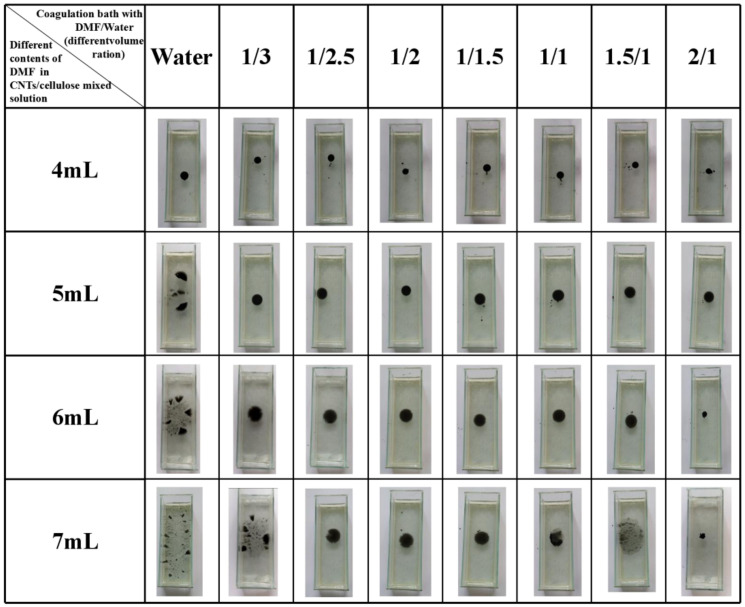
Digital photos of cellulose/CNTs mixed polymer solution added different DMF contents dropped into coagulating bath with DMF/water (different volume ratio).

**Figure 3 membranes-12-00369-f003:**
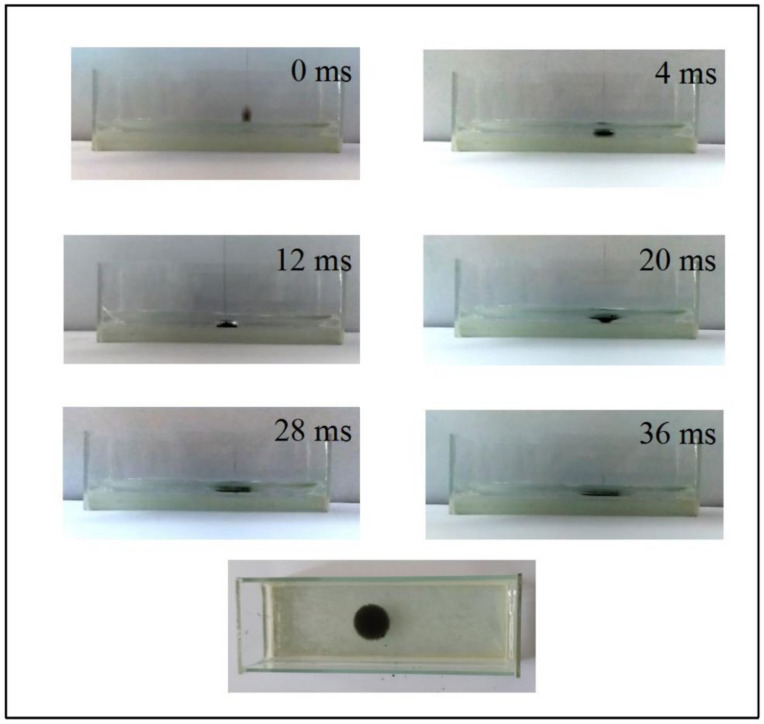
Digital photos of cellulose/CNTs mixed polymer solution added 6 mL DMF dropped into coagulating bath with DMF/water (volume ratio = 1/1) taken by high-speed camera.

**Figure 4 membranes-12-00369-f004:**
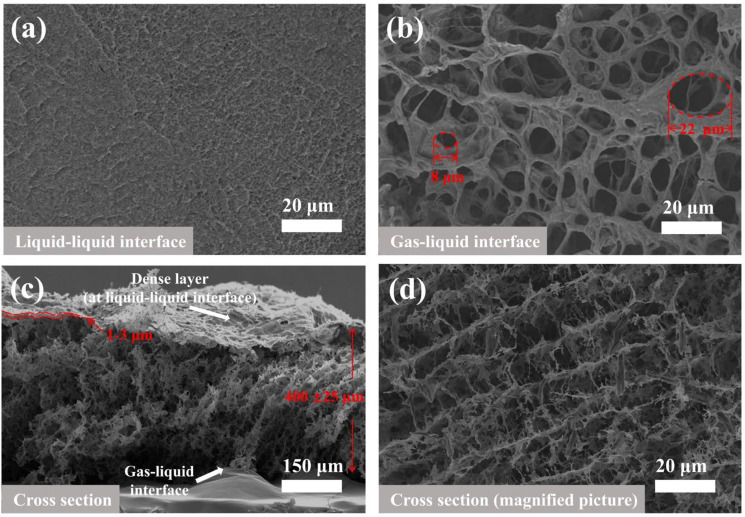
SEM images of (**a**) liquid–liquid interface; (**b**) gas–liquid interface; (**c**) cross section; (**d**) magnified cross section of the CCM.

**Figure 5 membranes-12-00369-f005:**
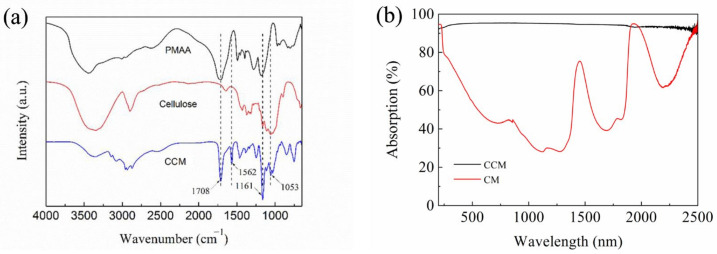
(**a**) FTIR spectra of the cellulose, PMAA and CCM; (**b**) UV-vis-NIR spectra of CM and CCM.

**Figure 6 membranes-12-00369-f006:**
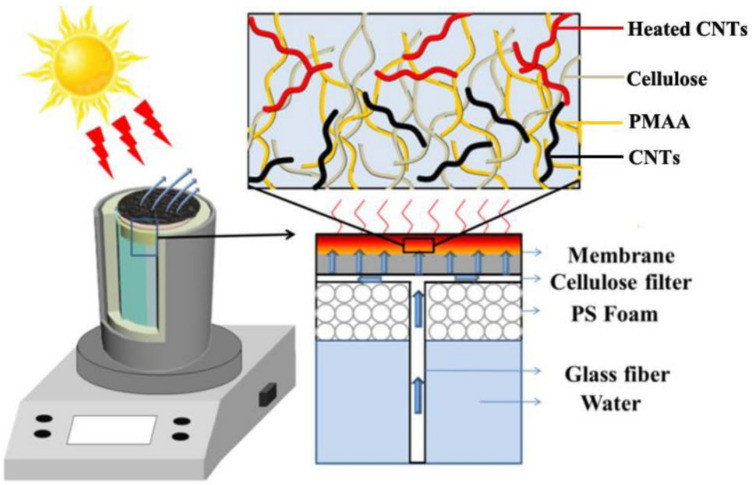
Schematic diagram of the transporter-assisted evaporation system.

**Figure 7 membranes-12-00369-f007:**
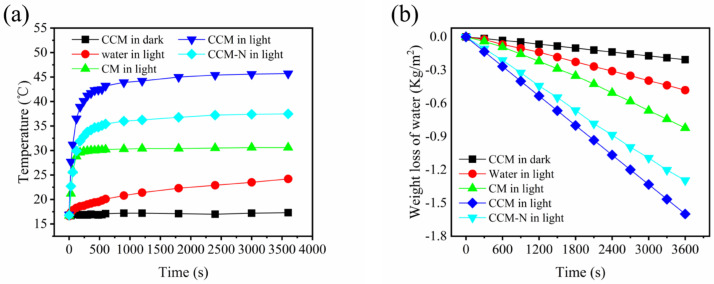
(**a**) Temperature changes of the water surface and membrane surface using CCM and CM under light and dark conditions; (**b**) weight loss of water using CCM and CM during the interfacial evaporation process under light and dark conditions.

**Figure 8 membranes-12-00369-f008:**
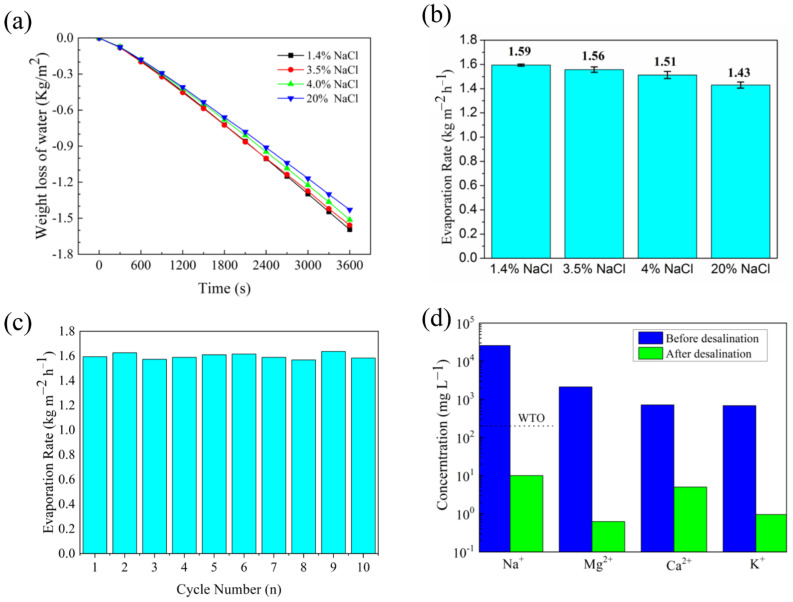
(**a**) Weight loss of water using the CCM when processing NaCl solutions with different salinities; (**b**) the corresponding evaporation rate of the CCM when processing NaCl solutions with different salinities; (**c**) stability test of the evaporation system using the CCM in 3.5 wt% NaCl solution; (**d**) ion concentrations using CCM before and after the solar-driven interfacial desalination.

## Data Availability

Not applicable.

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
