# Peer review of "Asymmetric Cellulose/Carbon Nanotubes Membrane with Interconnected Pores Fabricated by Droplet Method for Solar-Driven Interfacial Evaporation and Desalination"

_membranes, 2022, doi:10.3390/membranes12040369_

Round 1

Reviewer 1 Report

In this manuscript, the authors designed a cellulose/carbon nanotubes membrane with the high evaporation rate and energy efficiency of pure water and used it for solar-driven desalination. The work is interesting, and the writing is good. I suggest a minor revision before publication in Membranes. See detail below,

  1. The current title of this manuscript “Cellulose/carbon nanotubes membrane with interconnected pores fabricated by droplet method for solar-driven interfacial evaporation and desalination” cannot highlight the novelty of this work. I suggest revising it.
  2. In Figure 8, the authors provided the ion concentrations using CCM before and after the solar-driven interfacial desalination. To understand and evaluate its application performance, the desalination efficiency for different ions should be provided. For the calculation methods, the papers may helpful, such as Membranes2020,10, 280; Journal of Membrane Science, 2021, 618, 118668, Desalination, 2022, 523, 115394, etc.

Reviewer 2 Report

This work proposed a porous photothermal membrane prepared by droplet method. They discussed the formation mechanism of CCM and the photothermal desalination performance. The porous structure and its formation process have been emphasized in this work. However, the effect of the porous structure on the photothermal performance did not elucidate completely, which should be one of the highlights of this work.

  1. The Introduction should review the previous work about the preparation and structure of the photothermal membranes. A further description of droplet method is needed.
  2. The whole manuscript should be extensively polished.  
  3. What is the effect of the porous structure on the photothermal evaporation? I suggest that a control experiment on a non-porous cellulose/CNT membrane can be conducted.
  4. The surface tension is an important parameter to form a membrane in this work. I think photo data is not enough to describe the formation mechanism. The data of the surface tension should be measured by a surface tension meter.

Reviewer 3 Report

Comments and Suggestions for Authors

This paper focuses on developing a cellulose-carbon nanotubes based photothermal membrane using droplet method with application in solar driven interfacial evaporation and desalination.

The synthesized membrane has been characterized using techniques like Scanning electron microscopy (surface and cross section morphology) and ATR-FTIR (surface chemical composition). Finally, the authors have demonstrated through results the application of the membrane as a photothermal membrane absorber in interfacial evaporation systems and for water desalination.

Overall, the paper is well-written and easy to follow. The research topic is relevant today and extensive research has been carried out by the authors to bring together this paper. I feel it is worthy of publication in the Membranes journal.

I have listed down a few minor comments/suggestions which the author may consider for incorporating into the paper or commenting to make it even more comprehensive:

  1. Include more recent references: In the introduction, methods and discussion section include more recent and relevant papers from last 5-7 years to support your arguments and reasonings throughout the paper. For example:

Zhu, R., Wang, D., Liu, Y., Liu, M., & Fu, S. (2022). Bifunctional superwetting carbon nanotubes/cellulose composite membrane for solar desalination and oily seawater purification. Chemical Engineering Journal433, 133510.

Wang, Y., Qi, Q., Fan, J., Wang, W., & Yu, D. (2021). Simple and robust MXene/carbon nanotubes/cotton fabrics for textile wastewater purification via solar-driven interfacial water evaporation. Separation and Purification Technology254, 117615.

Liu, Xinghang, et al. "Towards highly efficient solar-driven interfacial evaporation for desalination." Journal of Materials Chemistry A 8.35 (2020): 17907-17937.

  1. Page 2, Material and Methods section: In the Material and methods section mention the membrane size (area) and volume of water used for the desalination studies.

  1. Page 7, Figure 4: If possible, include surface and cross-section SEM images of the control membrane (Cellulose membranes with no CNT’s). It would be an interesting piece of information for the reader and will give the reader a better understanding on how incorporation of carbon nanotubes changes the surface and cross-section morphology of the membrane.

  1. Page 7, Figure 4(c): If possible, please use a better-quality image instead. Also, use color to highlight or mark the dense layer, thickness, and the pore size of the membrane.

  1. Carbon nanotube loading in the membrane: Is the carbon nanotube load in the Cellulose-CNT membrane used in this paper optimized? Will it have any effect on membrane performance? Is this the membrane synthesis technique scalable to make bigger membranes?

Round 2

Reviewer 2 Report

The Introduction should be further revised in terms of logicality. 1) What's the preparation method of the designed membrane in paragragh 2? 2) The typical process of droplet method should be introduced, which is the key point. 3) It is strange to introduce the NIPS method at the end of paragraph 3. Besides, the English language should be polished carefully.
